# The Inactivation of the Putative Two-Component System Sensor PA14_27940 Increases the Susceptibility to Several Antibiotics and Reduces the Motility of *Pseudomonas aeruginosa*

**DOI:** 10.3390/ijms242417355

**Published:** 2023-12-11

**Authors:** Roberta Genova, Teresa Gil-Gil, Trinidad Cuesta, José Luis Martínez, Fernando Sanz-García

**Affiliations:** 1Centro Nacional de Biotecnología, Consejo Superior de Investigaciones Científicas, Darwin 3, 28049 Madrid, Spain; roberta.genova@eez.csic.es (R.G.); teresaggil94@gmail.com (T.G.-G.); tcuesta@cnb.csic.es (T.C.); 2Department of Biotechnology and Environmental Protection, Estación Experimental del Zaidín, Consejo Superior de Investigaciones Científicas, Prof. Albareda 1, 18008 Granada, Spain; 3EcLF Laboratory, Department of Biology, Emory University, Atlanta, GA 30322, USA; 4Departamento de Microbiología, Medicina Preventiva y Salud Pública, Universidad de Zaragoza, Domingo Miral sn, 50009 Zaragoza, Spain

**Keywords:** *Pseudomonas aeruginosa*, antibiotic resistance, MexXY, adjuvants, aminoglycosides, fosfomycin, swimming, swarming

## Abstract

The identification of targets whose inactivation increases the activity of antibiotics helps to fight antibiotic resistance. Previous work showed that a transposon-insertion mutant in the gene *PA14_27940* increases *Pseudomonas aeruginosa* susceptibility to aminoglycosides. Since polar effects may affect the phenotype, in the present work, we generated an in-frame *PA14_27940* deletion mutant. A *PA14_27940* deletion increased the susceptibility to aminoglycosides, tetracycline, tigecycline, erythromycin and fosfomycin. Excepting fosfomycin, the other antibiotics are inducers of the MexXY efflux pump. MexXY induction is required for *P. aeruginosa* resistance to these antibiotics, which is post-transcriptionally regulated by the anti-repressor ArmZ. Although *mexXY* is inducible by tobramycin in Δ*PA14_27940*, the induction level is lower than in the parental PA14 strain. Additionally, *armZ* is induced by tobramycin in PA14 and not in Δ*PA14_27940*, supporting that Δ*PA14_27940* presents an ArmZ-mediated defect in *mexXY* induction. For its part, hypersusceptibility to fosfomycin may be due to a reduced expression of *nagZ* and *agmK*, which encode enzymes of the peptidoglycan recycling pathway. Δ*PA14_27940* also presents defects in motility, an element with relevance in *P. aeruginosa*’s virulence. Overall, our results support that PA14_27940 is a good target for the search of adjuvants that will increase the activity of antibiotics and reduce the virulence of *P. aeruginosa*.

## 1. Introduction

*Pseudomonas aeruginosa* is a relevant opportunistic pathogen that presents a remarkable ability to infect immunocompromised hosts and a broad capacity of adaptation to different environments thanks to its high metabolic versatility [1,2,3,4,5]. It causes a wide range of infections, including complicated urinary tract or wound infections in hospitalized patients, as well as lower respiratory tract infections [6,7]. In addition, *P. aeruginosa* has been included in two groups of bacteria: ESKAPE (acronym of *Enterococcus faecium*, *Staphylococcus aureus*, *Klebsiella pneumoniae*, *Acinetobacter baumannii*, *P. aeruginosa* and *Enterobacter* spp.) and TOTEM (TOp Ten-resistant microorganisms), due to its impact on human health and its relevance in antibiotic resistance [8,9]. In fact, the treatment of *P. aeruginosa* is especially difficult due to its ability to easily and rapidly acquire resistance to multiple classes of antibiotics [3]. Aside from this acquired resistance, *P. aeruginosa* also exhibits a high intrinsic resistance to diverse antimicrobial agents, thanks to its restricted outer membrane permeability or its multidrug efflux pumps, among other mechanisms [3]. One of these efflux systems, MexXY-OprM, contributes to *P. aeruginosa* intrinsic resistance to different antimicrobial agents [10], including aminoglycosides, tetracycline and erythromycin [11]. The expression of this multidrug efflux pump is induced when *P. aeruginosa* is exposed to agents that cause ribosome stress [12]. This induction is a consequence of post-transcriptional regulation. Under these stress conditions, the expression of *armZ* is triggered (Figure 1). This gene encodes an anti-repressor that binds and sequesters MexZ—the main transcriptional repressor of *meXY*—hence allowing the induction of *mexXY* expression [13]. Additionally, *P. aeruginosa* also presents different mechanisms of intrinsic resistance to fosfomycin (Figure 1), including *fosA*, which encodes a fosfomycin inactivating enzyme [14], and the genes coding for enzymes in the alternative peptidoglycan recycling pathway, like *nagZ* or *agmK* [15,16].

The pathogenicity and success of *P. aeruginosa* as a human pathogen are attributed not only to its ability to develop antibiotic resistance to several classes of antibiotics, but also to the presence in its genome of genes encoding an arsenal of virulence factors [17]. Among the various virulence factors that are responsible for the pathogenicity of this bacterium, biofilm formation, increased motility—swarming or swimming—and the production of pigments and/or proteases have a pivotal role. It is worth mentioning that *P. aeruginosa*’ s motility plays a key role in the bacterial colonization of surfaces and biofilm formation; moreover, it has been observed that some bacteria, including *P. aeruginosa*, increase their resistance to some antibiotics when they are in the active swarming state [18]. On its hand, swimming favors the spread of bacteria in fluid environments and the colonization of host tissues [19].

Among the mechanisms that regulate antibiotic resistance and virulence phenotypes in *P. aeruginosa*, the two-component systems hold a substantial role [20,21]. Indeed, some two-component systems have been described to be implicated in both phenotypes, as ParRS, PmrAB or PhoPQ, which are involved in pathogenesis, virulence, and polymyxin and aminoglycoside resistance [22,23,24,25].

Previous work has shown that a transposon tagged mutant inactivating the two-component sensor encoded by *PA14_27940 (PA2798* in PAO1) presented an enhanced susceptibility to multiple aminoglycosides and an impaired virulence in *P. aeruginosa* PA14 strain [26,27]. *PA14_27940* is part of a two-gene operon that also includes *PA14_27950*, which codes for a putative anti-anti-sigma factor whose activity is dictated by its phosphorylation state (inactive upon phosphorylation). *PA14_27940* encodes a putative sensor phosphatase that activates the anti-anti-sigma factor encoded by *PA14_27950*. Since a transposon insertion mutant of *PA14_27940* showed an increased susceptibility to aminoglycosides and a reduced virulence [26], in the current work, we studied in detail these phenotypes and the potential mechanisms behind them. To carry this out, given that polar effects may affect the phenotype of the insertion mutant, we generated and analyzed an in-frame deletion mutant of *PA14_27940*. Our results support that the deletion of *PA14_27940* reduces the motility and increases the susceptibility to several antibiotics of *P. aeruginosa*. Therefore, this regulator may be a good target to search for compounds that simultaneously increase the activity of different antibiotics against *P. aeruginosa* and reduce the motility—and hence virulence—of this pathogen.

## 2. Results and Discussion

### 2.1. PA14_27940 Deletion Renders Hypersusceptibility to Several Antibiotics in P. aeruginosa

In previous work, various *P. aeruginosa* PA14 transposon insertion mutants were analyzed in order to find genes whose inactivation simultaneously resulted in a high susceptibility to drugs and a reduced expression of virulence factors, thus being potential targets of antivirulence/antiresistance co-adjuvants against this pathogen [26]. Among them, the gene *PA14_27940* was chosen to be further investigated because its corresponding insertion mutant displayed a phenotype of hypersusceptibility to aminoglycosides, as well as a reduced expression of virulence determinants [26]. Since insertion mutants may present polar effects, a clean in-frame deletion of this gene was needed to deepen into its effects on these phenotypes. Hence, a deletion mutant of *PA14_27940* (from now on dubbed Δ*PA14_27940*) was obtained by homologous recombination (see Section 3). The genome of the mutant was sequenced and analyzed in comparison with the one of the PA14 strain used in this work and the GenBank *P. aeruginosa* UCBPP-PA14 reference chromosome (NC_008463.1). The analysis showed that the mutant only presented the deletion of the gene *PA14_27940* when compared with the parental strain, which gave us the green light to carry out further experiments on it.

Thus, to determine the effect of *PA14_27940* deletion on *P. aeruginosa* susceptibility to antibiotics, the susceptibility of the Δ*PA14_27940* mutant and the PA14 parental strain to tobramycin, streptomycin, amikacin, kanamycin, tigecycline, tetracycline, ciprofloxacin, ceftazidime, imipenem, aztreonam, fosfomycin, erythromycin, chloramphenicol, colistin and polymyxin B was analyzed by an E-test.

In agreement with a previous analysis [28], the mutant was considered as hypersusceptible or resistant to an antibiotic if its MIC presented at least a two-fold change with respect to the one of PA14. Taking this into consideration, Δ*PA14_27940* was more susceptible than PA14 to the aminoglycoside tobramycin, with a four-fold reduction in the MIC, streptomycin (two-fold reduction), amikacin (six-fold reduction) and kanamycin (five-fold reduction), as well as to tigeclycline (three-fold reduction), tetracycline (two-fold reduction), erythromycin (three-fold reduction) and fosfomycin (four-fold reduction) (Table 1). Hence, these results show that the deletion of *PA14_27940* renders *P. aeruginosa* hypersusceptible to aminoglycosides, including tobramycin and amikacin, which belong to the therapy against this pathogen [29], fosfomycin and erythromycin. This result supports that *PA14_27940* may be a good target for the search of adjuvants that would increase the activity of several antibiotics against *P. aeruginosa*.

### 2.2. PA14_27940 Deletion Alters the Motility of P. aeruginosa

*P. aeruginosa*’s motility plays a key role in the bacterial colonization of surfaces and biofilm formation, thus its reduction is usually associated with a reduced virulence phenotype [30]. The former work showed that a transposon insertion mutant in *PA14_27940* presented a lower motility than the parental strain [26]. Consequently, we analyzed the motility of the Δ*PA14_27940* mutant. As shown in Figure 2, the mutant showed an impaired swarming phenotype: the mutant’s motility diameter was 43.3 mm, while PA14′s was 66.7 mm. Swimming motility was affected in the mutant as well, both in the motility pattern—regular circle for the PA14 and splash for the Δ*PA14_27940* mutant—and the motility diameters—33 and 15 mm for the wild-type strain and the mutant, respectively (Figure 2). In light of these results, we can assert that the deletion of *PA14_27940* entails a reduced motility in *P. aeruginosa* PA14, which indicates that the regulator encoded by this gene is implicated, directly or indirectly, in this virulence trait, reinforcing its candidacy as a potential target of antivirulence co-adjuvants in antipseudomonal therapies.

### 2.3. Effect of PA14_27940 in the Production of P. aeruginosa Virulence Determinants

Pyocyanin production, elastase activity and biofilm formation are important virulence factors for the infective process of *P. aeruginosa* [32,33,34,35,36], whose expression was impaired in the transposon insertion mutant of the gene *PA14_27940* [26]. Hence, we evaluated these phenotypes in the Δ*PA14_27940* mutant, with regard to the PA14 strain. As it is shown in Figure 3, although some slight decrease was observed in the Δ*PA14_27940* mutant for all virulence features, these differences were not statistically significant. Ergo, our results support that the clean deletion of *PA14_27940* does not affect either pyocyanin, elastase or biofilm production. Hence, this regulator is not likely to be involved in these phenotypes, in spite of our data obtained by analyzing the insertion mutant of this gene [26].

The analysis of the phenotype of insertion mutants is a valuable tool for global genetic analyses, and it has historically provided relevant results in the study of the bacterial intrinsic resistome [37]. However, our results support that, on occasion, the observed phenotype is not a direct consequence of the gene in question; in contrast, other features—such as polar effects—might compromise the phenotype. Therefore, our data strongly suggest that an ulterior analysis of phenotypes of interest in insertion mutants should require the construction of clean in-frame deletion mutants of the gene in question, before jumping to conclusions regarding said phenotypes.

### 2.4. The Loss of PA14_27940 Impedes the Induction of armZ, Hence Reducing the Induction of mexXY by Tobramycin

As stated above, the Δ*PA14_27940* mutant presents increased susceptibility to aminoglycosides, erythromycin, tigecycline and tetracycline, among other drugs, while susceptibility to beta-lactams and quinolones was not altered (Table 1). This phenotype is similar to that observed in a MexXY deficient mutant [10]. Since *mexXY* expression is induced by its antibiotic substrates, and this induction is required for the intrinsic resistance of *P. aeruginosa* to aminoglycosides, we measured the level of expression of *mexX* in the presence of subinhibitory concentrations of tobramycin in the Δ*PA14_27940* mutant and in the PA14 parental strain. As shown in Figure 4, although *mexX* expression was induced in both strains, the level of induction was significantly lower (*p* < 0.05) in the Δ*PA14_27940* mutant than in the parental PA14 strain. Albeit this lower induction might contribute to the increased susceptibility to aminoglycosides of Δ*PA14_27940*, the fact that *mexX* is still inducible suggests that other elements, besides *mexX* induction, might contribute to the hypersusceptibility phenotype.

As stated above, the anti-repressor ArmZ is a critical element in *mexXY* induction of the expression by the substrates of the efflux pump and hence in *P. aeruginosa* intrinsic resistance to aminoglycosides [10,13]. Consequently, we measured its expression in the presence of subinhibitory concentrations of tobramycin in the wild-type PA14 strain and in the Δ*PA14_27940* mutant.

As shown in Figure 5, subinhibitory concentrations of tobramycin induced *armZ* in the wild-type strain, while the expression of this gene was not induced in the Δ*PA14_27940* mutant, with the differences being statistically significant. Under these conditions, while MexZ—the repressor of the expression of *mexXY*—is sequestered by ArmZ in the wild-type PA14 strain [38], thus allowing the induction of *mexXY* expression in the presence of tobramycin, and the lack of *armZ* induction in the Δ*PA14_27940* mutant keeps the level of MexZ, and as a consequence, the induction of the expression of *mexXY* is hampered. Ergo, these results could explain the hypersusceptibility phenotype of the mutant.

### 2.5. Fosfomycin Hypersusceptibility of the ΔPA14_27940 Mutant Is Due to the Reduced Expression of the Genes Encoding the Alternative Peptidoglycan Recycling Pathway

As shown above in Table 1, the Δ*PA14_27940* mutant presents an increased susceptibility to fosfomycin. Since this drug has been proposed as last-resort antipseudomonal therapy [31] and the hypersusceptibility of the mutant to fosfomycin cannot be explained by the lack of *mexXY* induction—fosfomycin is not a MexXY substrate—we aimed to separately delve into the mechanisms behind this phenotype. In this sense, it is worth noting that *P. aeruginosa* presents different mechanisms of intrinsic resistance to fosfomycin (Figure 1), including the presence in its genome of *fosA*, which encodes a fosfomycin inactivating enzyme and the genes coding for enzymes in the alternative peptidoglycan recycling pathway [14,15,39,40]. Therefore, this increased susceptibility could be a consequence of an increase in the intracellular concentration of fosfomycin or a reduced expression of the genes encoding for the alternative peptidoglycan recycling proteins. To address the first possibility, the intracellular accumulation of fosfomycin was measured in PA14 and in the Δ*PA14_27940* mutant. As shown in Figure 6, no differences were found between both strains, supporting that an increased accumulation of fosfomycin was not the cause of the hypersusceptibility to this antibiotic displayed by the Δ*PA14_27940* mutant.

Since the amount of intracellular fosfomycin was not significantly lower in the *PA14_27940* mutant than in PA14, the de novo synthesis pathway would be similarly inhibited in both strains. In conditions when this pathway is inhibited, the recycling pathway is a salvage mechanism that allows the formation of the peptidoglycan, hence reducing the fosfomycin activity (Figure 1). Consequently, a reduction in the expression of the enzymes of the peptidoglycan recycling pathway will produce an increased susceptibility to fosfomycin as observed in the *PA14_27940* mutant. To analyze this possibility, the expression of the genes that code for the enzymes of the alternative peptidoglycan recycling pathway, i.e., *anmK*, *agmK* and *nagZ*, was measured in PA14 and in the *PA14_27940* mutant.

As shown in Figure 7, the expression of *anmK* and *nagZ* was significantly lower in the mutant lacking *PA14_27940* than in PA14. These results indicate that a decreased expression of the enzymes of the alternative peptidoglycan recycling pathway might be on the basis of the phenotype of fosfomycin hypersusceptibility displayed by the *PA14_27940* mutant, thus suggesting a potential nexus between this two-component sensor and those genes involved in fosfomycin intrinsic resistance, a feature that would require further research.

## 3. Materials and Methods

### 3.1. Bacterial Strains, Plasmids and Oligonucleotides

The strains, plasmids and oligonucleotides used in this study are described in Table 2 and Table 3.

### 3.2. Bacterial Growth Conditions

Unless otherwise stated, all strains were grown in rich Luria Bertani Broth (LBB) Lennox (Conda, Torrejón de Ardoz, Spain) at 37 °C and 250 rpm or in Luria Bertani Agar (LBA) (LBB with 1.5% agar (Conda, Torrejón de Ardoz, Spain)) at 37 °C. Different antibiotics were added to the growth media when required: 100 μg/mL ampicillin for the selection of *E. coli* DH5α and *E. coli* S17-1 clones containing pGEM^®^-T Easy or RGD001, respectively, and 350 μg/mL carbenicillin for the selection of *P. aeruginosa* clones containing the deletion of *PA14_27940* [42].

### 3.3. Construction of ΔPA14_27940 P. aeruginosa Mutant

The deletion of the gene *PA14_27940* in *P. aeruginosa* PA14 strain was performed by homologous recombination as herein described. An insert that comprised 500 bp upstream and 500 bp downstream of the gene, flanked by HindIII restriction sites, was obtained by PCR using the primers PA14_27940.ups_fw, PA14_27940.ups_rv, PA14_27940.dns_fw and PA14_27940.dns_rv (Table 3). Then, the PCR product was introduced in a HindIII-digested and dephosphorylated pEX18Ap vector [42] by using T4 DNA ligase (New England Biolabs, NEB, Ipswich, MA, USA) and incubating O/N at 16 °C. The resultant plasmid, RGD001, was introduced by transformation in *E. coli* S17-1, a donor strain that was submitted to conjugation and mutant selection with *P. aeruginosa* PA14 strain, as described elsewhere [42], using 350 μg/mL of carbenicillin and 10% of sucrose to select the deletion-containing bacteria. The deletion of the gene was verified by PCR with primers PA14_27940.comp_fw-PA14_27940.comp_rv and PA14_27940.ups_fw-PA14_27940.dns_rv (Table 3).

### 3.4. Whole-Genome Sequencing and Bioinformatics Analysis 

Genomic DNA extractions and DNA quality check were performed by the Translational Genomics Unit, Instituto Ramón y Cajal de Investigación Sanitaria (IRYCIS), Hospital Ramón y Cajal (HRYC), Madrid, Spain. Genomic DNA of each population was extracted by Chemagic™ DNA Bacterial Kit H96 (CMG-799 Chemagic™), using the equipment Chemagic™ 360/MSMI (PerkinElmer, Waltham, MA, USA), and DNA quality check was performed by Agilent 2200 TapeStation System bioanalyzer (Santa Clara, CA, USA). The whole-genome sequencing (WGS) was performed by the Oxford Genomics Centre. Constructed libraries were paired-end (2 × 350) and sequenced with NovaSeq6000 Illumina system (San Diego, CA, USA). The average number of reads per sample represented a coverage of 300×. WGS data were analyzed using the CLC Genomics Workbench 21.0.5 software (QIAGEN, Hilden, Germany). Genomic information was trimmed, and the reads were aligned against the GenBank *P. aeruginosa* UCBPP-PA14 reference chromosome (NC_008463.1). Δ*PA14_27940* mutant’s genome was also filtered against the sequence of the PA14 strain used in our laboratory. The genome is accessible with the BioProject database ID code PRJNA1012553.

### 3.5. Analysis of Susceptibility to Antibiotics 

The susceptibility of the different strains to tobramycin, streptomycin, amikacin, kanamycin, tigecycline, tetracycline, ciprofloxacin, ceftazidime, imipenem, aztreonam, fosfomycin, erythromycin and chloramphenicol was determined using antibiotic gradient strips (MIC Test Strip, Liofilchem^®^, Roseto degli Abruzzi, Italy) in Mueller Hinton Agar (MHA) (Sigma, St. Louis, MO, USA), whereas susceptibility to colistin and polymyxin B was determined in MHA II. Plates were incubated 24 h at 37 °C to read the MIC values.

### 3.6. Motility Assays

Motility assays were carried out in fresh agar plates containing 25 mL of medium. Swarming assays were performed in a medium that contained 0.5% Casamino Acids, 0.5% Bacto agar, 0.5% filtered glucose, 3.3 mM K_2_HPO_4_ and 3 mM MgSO_4_, as formerly depicted [43]. Swimming assays were made in 20 g/l LBB with 0.3% agar [44]; in a medium containing 1% tryptone, 0.5% yeast extract and 0.5% NaCl. A 4 µL inoculum (OD600 = 1) of the bacterial strains was placed on the center of the agar surface. Three replicates of each strain were incubated for 24 h at 37 °C. After the incubation, the motility zone of the swarming and swimming assays was measured.

### 3.7. Elastase Activity and Pyocyanin Production

To measure elastase activity and pyocyanin production, the bacterial strains were first cultured in 10 mL of LBB at 37 °C for 24 h. After incubation, 1 mL samples were collected, centrifuged for 10 min at 7000 rpm, and the supernatants were filtered through 0.2 µm pore-size filters (Whatman, Maidstone, UK). The elastase assay was performed as follows: 100 µL of each supernatant was incubated for 2 h with 1 mL of 5 mg/mL of Congo red elastin (Sigma-Aldrich, St. Louis, MO, USA) in TrisHCl 100 mM (pH = 7.5) and CaCl 1 mM, at 37 °C and 250 rpm. Later, samples were centrifuged for 10 min at 13,000 rpm. A total of 100 µL of each supernatant was placed in a 96-well microtiter plate (Nunc) and the elastase activity was quantified by reading the optical density at 495 nm (OD_495_) in a Tecan Spark 10 M plate reader (Tecan, Mennedorf, Switzerland). Two technical and three biological replicates were included in the assay. On the other side, the pyocyanin production was determined by placing 100 µL of the above-said filtered supernatants in a 96-well microtiter plate (Nunc) and measuring the OD_690_, using the same plate reader. Three biological replicates of each strain were included.

### 3.8. Biofilm Formation

The study of biofilm formation was performed using 96-well microtiter plates (Falcon 3911 Microtest III flexible assay plate (Fisher, Hampton, VA, USA)) as previously described [26,45]. Briefly, each strain was grown at 37 °C for 24 h and 250 rpm in LBB. The O/N bacterial cultures were diluted 1:100 and inoculated into the microtiter plate (100 µL/well). The plate was kept at 37 °C for 48 h. Next, each well was incubated with 25 µL of a solution of 0.1% crystal violet in ethanol for 5 min, and the excess dye was washed with distilled water for 4 times. After, in order to detach the biofilm from the wells, 150 µL of Triton X-100 (0.25%) was added to each well and 100 µL of each sample was transferred to a 96-well microtiter plate (Nunc, Roskilde, Denmark). The quantification of biofilm formation was performed by measuring the OD_570_ in a Tecan Spark 10 M plate reader (Tecan).

### 3.9. Quantification of Intracellular Fosfomycin

Assays to test the intracellular concentration of fosfomycin in bacterial cells were conducted using a bioassay as previously stated [46]. *P. aeruginosa* PA14 wild-type strain, as well as *P. aeruginosa* Δ*glpT* and Δ*fosA*, were used as controls. Briefly, 10^9^ cells/mL bacterial suspensions containing 2 mg/mL of fosfomycin were incubated at 37 °C (60 min) and washed sequentially three times in a buffer containing 10 mM Tris [pH 7.3], 0.5 mM MgCl_2_ and 150 mM NaCl to remove the remaining extracellular Fosfomycin. After washing, cells were suspended in 0.6 mL of 0.85% NaCl. The number of CFU/mL in each sample was determined by plating dilutions of the suspensions onto LB agar that were incubated 20 h at 37 °C. The samples were boiled for 5 min at 100 °C to release the intracellular fosfomycin, and cellular debris were removed by centrifugation (11,900 g, 10 min). A total of 40 μL of each supernatant was poured on sterilized disks (9 mm; Macherey-Nagel, Düren, Germany), and the disks were deposited onto LB agar plates, previously overlaid with OmniMAX *E. coli*. In parallel, disks containing different amounts of fosfomycin were also used in the same conditions to make a titration curve. The plates were incubated for 20 h at 37 °C. The diameter of inhibition zones was measured and compared with the ones of the controls containing different amounts of fosfomycin to quantify the concentration of this antibiotic in each sample. The fosfomycin concentration was represented as micrograms per 10^7^ cells.

### 3.10. RNA Extraction and qRT-PCR

RNA was extracted from three cultures of each strain. First, 20 mL of LBB medium were inoculated from O/N cultures to a final OD_600_ of 0.01 and grown until exponential phase (OD_600_ = 0.6) was reached, in presence and absence of a subinhibitory concentration of tobramycin (1/8 of MIC of each strain). Then, the cultures were spun down, and the resulting pellets were collected and used to perform RNA extraction, following the protocol of RNeasy mini Kit (QIAGEN). The residual DNA was removed by two different DNA treatments: the first one with DNase (QIAGEN) and the second one with Turbo DNA-free (Ambion, Seoul, Republic of Korea). The absence of DNA was checked by PCR, using primers rplU_fw and rplU_rv (Table 3). After, cDNA was obtained from 2.5 to 5 μg of RNA in a final volume of 20 μL, using the High-Capacity cDNA reverse transcription kit (Applied Biosystems, Waltham, MA, USA). qRT-PCR was performed in an ABI Prism 7300 Real-time PCR system (Applied Biosystems), using Power SYBR green PCR master mix (Applied Biosystems). A total of 50 ng of cDNA was used for the reaction. Primer3 Input software (https://bioinfo.ut.ee/primer3-0.4.0/primer3/, access date 6 June 2023) was utilized to design the primers (Table 3), and their efficiency was analyzed by qRT-PCR using serial dilutions of cDNA. Said primers were used at a concentration of 10 μM. All mRNA expression values were determined as the average of three independent biological replicates, each one with three technical replicates. Primers rplU_fw and rplU_rv were used to quantify the expression of the housekeeping gene *rplU* that was used to normalize the results, and the relative expression of each gene was calculated with the 2^−ΔΔCt^ method [47].

### 3.11. Statistical Analyses

Statistical analyses were performed with Microsoft Excel software vs 16.55 by using *t*-test.

## 4. Conclusions

Studies on antibiotic resistance and bacterial virulence are usually performed independently. However, different works have shown that these two clinically relevant phenotypes can be interlinked. Frequently, the acquisition of resistance produces a fitness cost that can be reflected in a lower virulence, but on other occasions, the expression of antibiotic resistance genes and virulence determinants is simultaneously regulated by global regulators. Two-component systems are critical elements for the adaptation of bacteria to grow in different ecosystems, because they sense the environmental conditions and respond accordingly. In this regard, it is not unexpected that two-component systems such as ParRS, PmrAB or PhoPQ are involved in the virulence and the susceptibility to antibiotics of *P. aeruginosa* [22,23,24,25]. In the current work, we show that the deletion of *PA14_27940*, which encodes the sensor of a two-component system, increases the susceptibility to several antibiotics and reduces the motility of *P. aeruginosa*. While we propose mechanisms underlying the increased antibiotic susceptibility, the reasons for the reduced motility of this mutant remain obscure and merit being analyzed in further studies. Together with previous results [22,23,24,25], this indicates that two-component systems, such as *PA14_27940–PA14_27950*, could be good targets in the search of adjuvants that simultaneously increase the activity of antibiotics and reduce the virulence of *P. aeruginosa*.

## Figures and Tables

**Figure 1 ijms-24-17355-f001:**
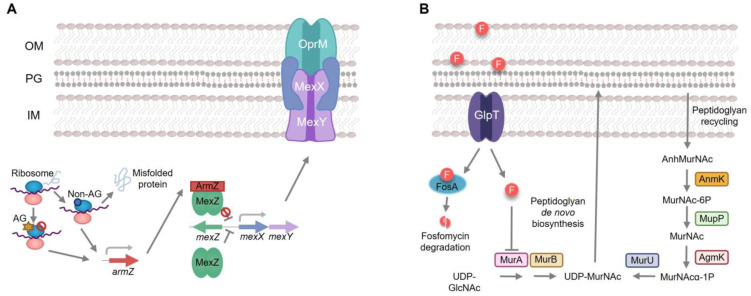
Intrinsic mechanisms of resistance to aminoglycosides and fosfomycin in *P. aeruginosa*. (**A**). MexXY is a *P. aeruginosa* efflux pump able to extrude aminoglycosides. Its expression is regulated by the MexZ repressor. Aminoglycosides or other ribosome-damaging agents induce the expression of *armZ*, which encodes a MexZ anti-repressor. When *armZ* is induced, ArmZ sequesters MexZ and the expression of MexXY is triggered, hence giving rise to aminoglycoside resistance. (**B**). Fosfomycin enters in *P. aeruginosa* through the GlpT transporter and exerts its action by inhibiting MurA, the first enzyme of the de novo pathway of peptidoglycan synthesis. Mutations in the genes encoding these elements are the major causes of acquired resistance to fosfomycin in *P. aeruginosa*. In addition, *P. aeruginosa* produces FosA, an enzyme capable of degrading fosfomycin and presents a peptidoglycan recycling pathway capable of producing the peptidoglycan even when the de novo pathway is inhibited (c.a. in presence of fosfomycin). These two elements contribute to the intrinsic resistance to fosfomycin of *P. aeruginosa*. OM: Outer membrane, PG: peptidoglycan, IM: inner membrane, AG star: aminoglycosides, block symbol: inhibition.

**Figure 2 ijms-24-17355-f002:**
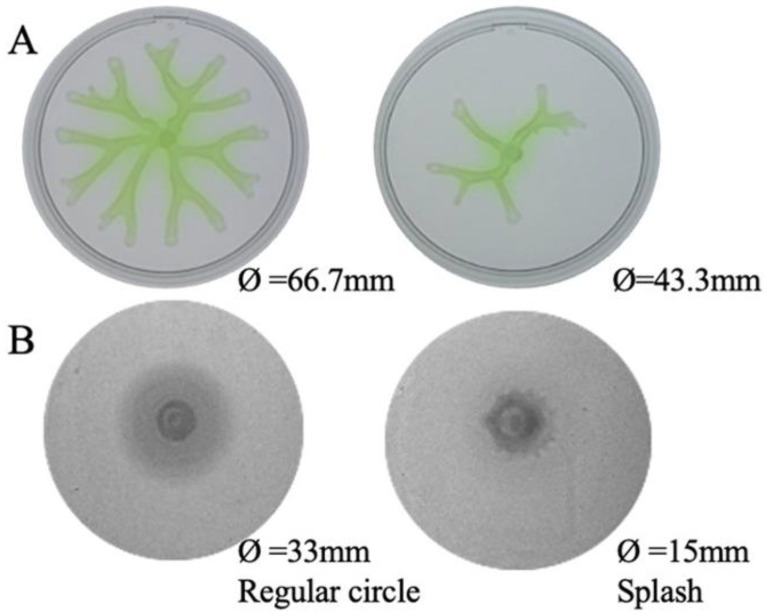
Motility phenotype of *P. aeruginosa* Δ*PA14_27940* mutant. The image illustrates the swarming (**A**) and swimming (**B**) phenotype of *P. aeruginosa* PA14 strain and Δ*PA14_27940* mutant. The plates are shown in the order: PA14 and Δ*PA14_27940*. The diameters of the strains are reported in the figure at the bottom of the images. The swimming shapes are identified following the classification included in [31]. Both swimming and swarming were statistically significantly lower in Δ*PA14_27940* mutant than in PA14 (*p* < 0.05).

**Figure 3 ijms-24-17355-f003:**
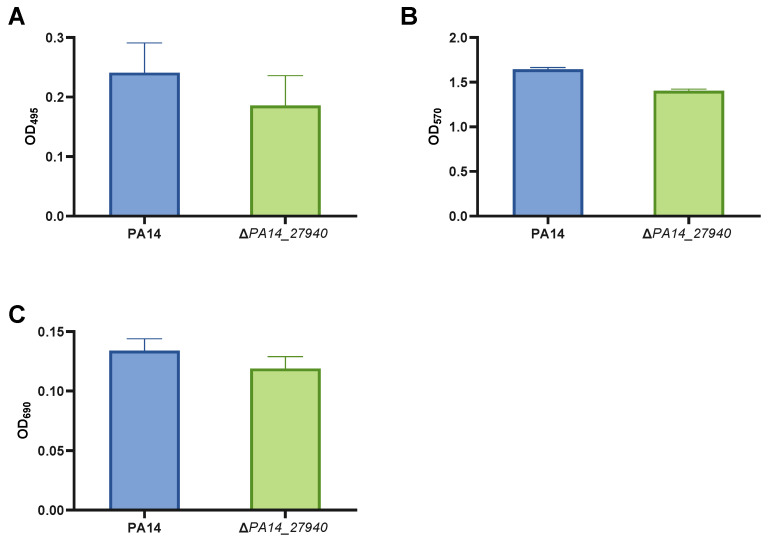
Quantification of virulence factors in *P. aeruginosa* Δ*PA14_27940* mutant. The figure exposes elastase activity data (**A**), biofilm formation assay data (**B**) and pyocyanin production data (**C**) of *P. aeruginosa* PA14 strain and Δ*PA14_27940* mutant. Error bars indicate standard deviations of the results from eight independent experiments in the biofilm formation assay and from three experiments in the other analysis. Differences between bars are not statistically significant (*p* > 0.05; by *t*-test, in all cases).

**Figure 4 ijms-24-17355-f004:**
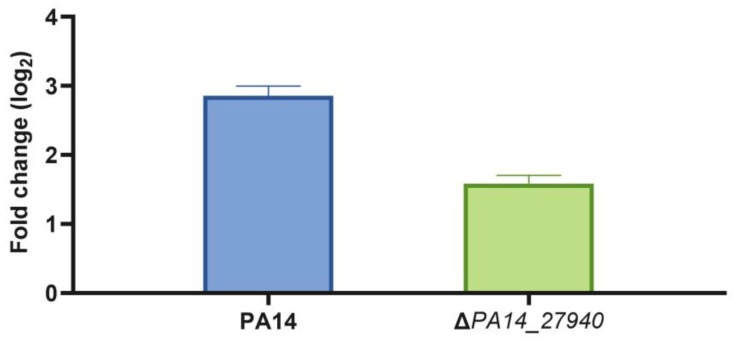
Induction of *mexX* by subinhibitory concentrations of tobramycin in *P. aeruginosa* Δ*PA14_27940* mutant. The level of *mexX* expression in PA14 and Δ*PA14_27940* was measured in presence of subinhibitory concentrations of tobramycin (1/8 of its respective MICs). Error bars indicate standard deviations of the results from three independent experiments. The induction level of the Δ*PA14_27940* mutant was statistically significantly lower than the one of PA14 (*p* < 0.05).

**Figure 5 ijms-24-17355-f005:**
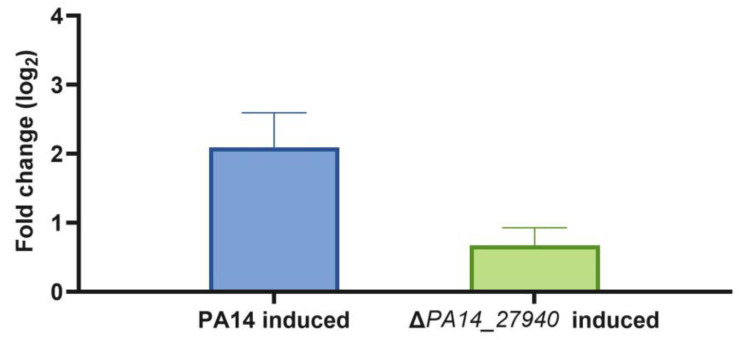
Induction of *armZ* by subinhibitory concentrations of tobramycin in *P. aeruginosa* Δ*PA14_27940* mutant. The level of *armZ* expression in PA14 and Δ*PA14_27940* was measured in presence of subinhibitory concentrations of tobramycin (1/8 of their respective MICs). Error bars indicate standard deviations of the results from three independent experiments. The induction level of the Δ*PA14_27940* mutant was statistically significantly lower than the one of PA14 (*p* < 0.05).

**Figure 6 ijms-24-17355-f006:**
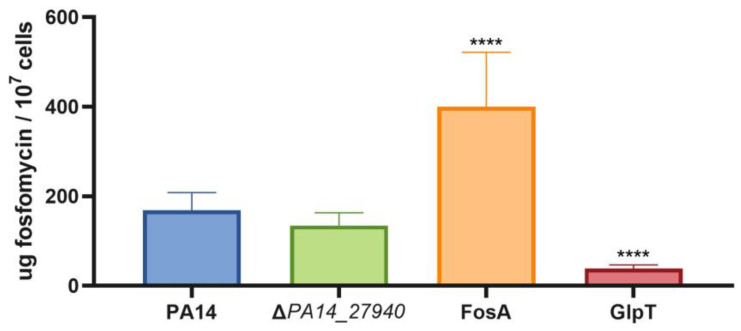
Intracellular accumulation of fosfomycin in *P. aeruginosa* Δ*PA14_27940* mutant. The figure shows the intracellular accumulation of fosfomycin by the wild-type PA14 strain and the Δ*PA14_27940* mutant. A mutant defective in FosA (a fosfomycin-inactivating enzyme) and another in GlpT (the fosfomycin transporter) were used as controls of increased and reduced fosfomycin accumulation, respectively. Error bars indicate standard deviations of the results from three independent experiments. Differences between bars are not statistically significant between PA14 and the Δ*PA14_27940* mutant: Statistically significance were calculated with *t*-test for paired samples assuming equal variances: **** *p* < 0.0005.

**Figure 7 ijms-24-17355-f007:**
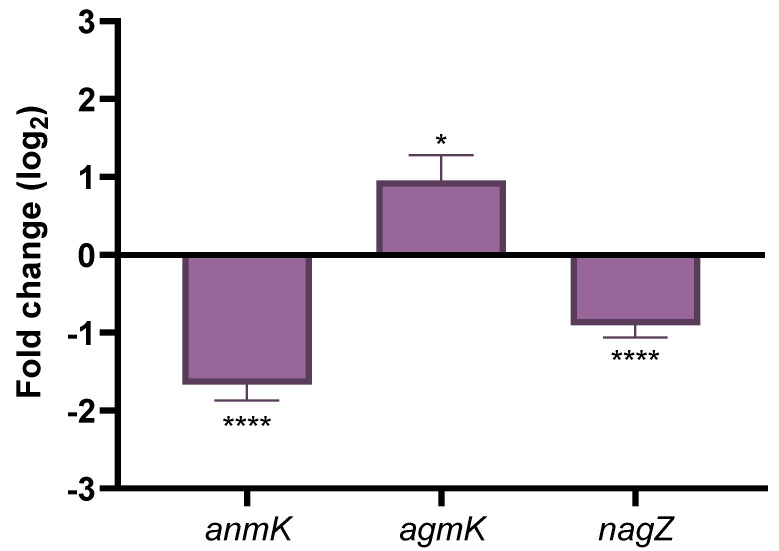
Expression of genes encoding the enzymes of the alternative peptidoglycan recycling pathway. The figure shows the fold-change of the expression of *anmK*, *agmK* and *nagZ* in Δ*PA14_27940* mutant as compared with PA14. Error bars indicate standard deviations of the results from three independent experiments. Statistically significant differences between PA14 and the Δ*PA14_27940* mutant were calculated with *t*-test for paired samples assuming equal variances: * *p* < 0.05, **** *p* < 0.0005.

**Table 1 ijms-24-17355-t001:** Susceptibility of *P. aeruginosa* Δ*PA14_27940* to different antibiotics.

Antibiotic	MIC (µg/mL)	
	PA14	Δ*PA14_27940*
TOB	0.75	0.19 *
STR	6	3 *
AMK	1.5	0.25 *
KAN	32	6 *
TGC	3	1 *
TET	24	12 *
CIP	0.032	0.032
CAZ	0.5	0.5
IPM	0.75	0.75
ATM	3	2
FOF	16	4 *
ERY	96	32 *
CHL	32	24
CS	1	1
PB	1	1

Tobramycin (TOB), streptomycin (STR), amikacin (AMK), kanamycin (KAN), tigecycline (TGC), tetracycline (TET), ciprofloxacin (CIP), ceftazidime (CAZ), imipenem (IPM), aztreonam (ATM), fosfomycin (FOF), erythromycin (ERY), chloramphenicol (CHL), colistin (CS) and polymyxin B (PB). Δ*PA14_27940* was considered as hypersusceptible to an antibiotic (indicated with an asterisk) if its MIC presented at least a two-fold change respect to the one of PA14.

**Table 2 ijms-24-17355-t002:** Bacterial strains used in this work.

Bacterial Strains	Description	Reference/Origin
*Escherichia coli* DH5α	Strain used for cloning	Laboratory collection
*E. coli* S17-1	Donor strain for conjugation	[41]
*Pseudomonas aeruginosa* PA14	Laboratory model strain of *P. aeruginosa*	Laboratory collection
Δ*PA14_27940*	Mutant of *P. aeruginosa* PA14 with *PA14_27940* deletion	This study

**Table 3 ijms-24-17355-t003:** Plasmids and oligonucleotides used in this work.

Plasmids	Description	Origin
pGEM^®^-T Easy	Commercial plasmid used for cloning optimization of PCR products	Promega (Madison, WI, USA)
pEX18Ap	Cloning vector	[42]
RGD001	pEX18Ap containing 500 bp upstream and 500 bp downstream of the gene *PA14_27940*	This study
**Oligonucleotides**	**Sequence 5′3′**	**Utilization**
PA14_27940.ups_fw	AAGCTGGGCTTCCAGTTCCAGGC	Mutant construction
PA14_27940.ups_rv	TTACCGGTACTCATGCAAGGCTTTATGCATGGGTCATCCA	Mutant construction
PA14_27940.dns_fw	TGGATGACCCATGCATAAAGCCTTGCATGAGTACCGGTAA	Mutant construction
PA14_27940.dns_rv	AAGCTAAGGGATCGACCGGTTAA	Mutant construction
PA14_27940.comp_fw	GGCCTGCAGATCTTCGAAAG	Mutant construction
PA14_27940.comp_rv	GTCACCGGAAGCATGTTCAT	Mutant construction
rplU_fw	CGCAGTGATTGTTACCGGTG	qRT-PCR
rplU_rv	AGGCCTGAATGCCGGTGATC	qRT-PCR
mexX_fw	GTACGAGGAAGGCCAGGAC	qRT-PCR
mexX_rv	CTTGATCAGGTCGGCGTAG	qRT-PCR
M13_fw	CGCCAGGGTTTTCCCAGTCACGAC	Sequencing
M13_rv	CAGGAAACAGCTATGAC	Sequencing
armZ_RT_F	ATCCTGCAAGAGCATGTCA	qRT-PCR
armZ_RT_R	GACGTCGAGCAGTTCCAG	qRT-PCR
agmK_RT_F	AGCTGAATCGCTGGTTGGAC	qRT-PCR
agmK_RT_R	AACGGTCGGCAGTCTTCCTG	qRT-PCR
anmK_RT_F	CAACGTGCTGATGGACGCCT	qRT-PCR
anmK_RT_R	AGCCAGGACAGGTTGAAGCG	qRT-PCR
nagZ_RT_F	AGGTGGGCGGGCTGATCATCTT	qRT-PCR
nagZ_RT_R	ATTGGGGTTGTCGGCGATCG	qRT-PCR

## Data Availability

Data is contained within the article.

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
