# Peer review of "The Inactivation of the Putative Two-Component System Sensor PA14_27940 Increases the Susceptibility to Several Antibiotics and Reduces the Motility of Pseudomonas aeruginosa"

_ijms, 2023, doi:10.3390/ijms242417355_

Round 1
Reviewer 1 Report
Comments and Suggestions for Authors
Authors stand as accomplished researchers, widely recognized for their contributions to this topic.
This article represents a natural progression from their earlier groundbreaking research, where they demonstrated the inactivation of the putative two-component system sensor 2 PA14_27940.
The manuscript's English quality is exemplary, a testament to the authors' dedication to precision and clarity.
What mechanisms are proposed for the increased susceptibility of the ΔPA14_27940 mutant to fosfomycin?
How was the intracellular accumulation of fosfomycin measured, and what were the results?
Generate a graph representing the results of the qRT-PCR experiments, showing the expression levels of genes in the presence and absence of tobramycin. Graphs can be effective in conveying trends and differences in gene expression.
Discuss any potential future directions suggested by the study's findings, especially in terms of developing therapeutic strategies or further investigating the role of PA14_27940 in P. aeruginosa biology. Evaluate how well the results are integrated into the broader context of antibiotic resistance, motility, and virulence in P. aeruginosa.
Author Response
We appreciate the positive opinion of the referee on our work.
Following referee's suggestions, the methods for measuring intracellular fosfomycin are described in full and a more detailed explanation of the causes of fosfomycin hypersusceptibility is included. To facilitate this explanation, a new figure has been added.
In addition, the conclusions section has been expanded following referee's suggestions.
Reviewer 2 Report
Comments and Suggestions for Authors
The author's results support that the deletion of PA14_27940 reduces the motility and increases the susceptibility to several antibiotics of P. aeruginosa. Therefore, this regulator may be a good target for searching for compounds that simultaneously increase the activity of different antibiotics against P. aeruginosa and reduce this pathogen's motility and virulence.
In the current manuscript, the Authors show that the deletion of PA14_27940, which encodes the sensor of a two-component system, increases the susceptibility to several antibiotics and reduces the motility of P. aeruginosa.
The authors have described each experiment in detail, each measurement is characterized with appropriate statistics.
Minor:
In chapter 3, in the statistical analysis, the authors should also mention the software they used for the T-test.
The authors could schematically show P. Aeruginosa with a multidrug efflux pump and molecular and biochemical objects associated with resistance, so that the scheme follows the introduction and discussion, which would make the text much easier to follow.
Author Response
We appreciate the positive opinion of the referee in our work.
Following referee's suggestions, a new Figure has been added and we have included in the methods section that T-text was performed with Excel
Reviewer 3 Report
Comments and Suggestions for Authors
This study investigates the gene PA14_27940 in Pseudomonas aeruginosa as a potential target to combat antibiotic resistance. Deleting PA14_27940 increased susceptibility to various antibiotics, indicating its role in resistance. The mutant also showed defects in MexXY efflux pump induction, likely mediated by the anti-repressor ArmZ. Additionally, the study identified heightened susceptibility to fosfomycin due to reduced expression of genes involved in peptidoglycan recycling. The ΔPA14_27940 mutant exhibited motility defects and reduced virulence. Overall, the findings suggest PA14_27940 as a promising target for developing adjuvants to enhance antibiotic efficacy and reduce P. aeruginosa virulence.It is a well written paper based on all relevant bibliography.Protocol well designed.Results are extensively discussed
The paper merits publication.
Author Response
We appreciate the positive opinion of the referee on our work
Round 2
Reviewer 1 Report
Comments and Suggestions for Authors
I do not have more comment. Thanks